# Type 2 diabetes remission trajectories and variation in risk of diabetes complications: A population-based cohort study

Hajira Dambha-Miller[1], Hilda O. Hounkpatin[1]*, Beth Stuart[1¤], Andrew Farmer[2], Simon Griffin[3,4]

1 Primary Care Research Centre, School of Primary Care Population Sciences and Medical Education, University of Southampton, Southampton, England, 2 Nuffield Department of Primary Care Health Sciences, University of Oxford, Oxford, England, 3 Department of Public Health and Primary Care, School of Clinical Medicine, University of Cambridge, Cambridge, England, 4 MRC Epidemiology Unit, School of Clinical Medicine, University of Cambridge, Cambridge, England

¤ Current address: Wolfson Institute of Population Health, Queen Mary University of London, London, England
* H.O.Hounkpatin@soton.ac.uk

**Data Availability Statement:** "We do not have governance permissions to share individual-level data on which these analyses were conducted since they derive from clinical record data.

## Abstract

Biochemical remission of type 2 diabetes is achievable through dietary changes, physical activity and subsequent weight loss. We aim to identify distinct diabetes remission trajectories in a large population-based cohort over seven-years follow-up and to examine associations between remission trajectories and diabetes complications. Group-based trajectory modelling examined longitudinal patterns of HbA$_{1c}$ level (adjusting for remission status) over time. Multivariable Cox models quantified the association between each remission trajectory and microvascular complications, macrovascular complications, cardiovascular (CVD) events and all-cause mortality. Four groups were assigned. Group 1 (8,112 [13.5%]; achieving HbA$_{1c}$ <48 mmol/mol (6.5%) followed by increasing HbA$_{1c}$ levels); Group 2 (6,369 [10.6%]; decreasing HbA$_{1c}$ levels >48 mmol/mol (6.5%)); Group 3 (36,557 [60.6%]; stable high HbA$_{1c}$ levels); Group 4 (9,249 [15.3%]; stable low HbA$_{1c}$ levels (<48mmol/mol or <6.5%)). Compared to Group 3, Groups 1 and 4 had lower risk of microvascular complications (aHRs (95% CI): 0.65 (0.61–0.70), p-value <0.001;0.59 (0.55–0.64) p-value<0.001, respectively)), macrovascular complications (aHRs (95% CI): 0.83 (0.75–0.92), p-value<0.001; 0.66 (0.61–0.71), p-value<0.001) and CVD events (aHRs (95% CI): 0.74 (0.67–0.83), p-value<0.001; 0.67(0.61–0.73), p-vlaue<0.001). Risk of CVD outcomes were similar for Groups 2 and 3. Compared to Group 3, Group 1 (aHR: 0.82(95% CI: 0.76–0.89)) had lower risk of mortality, but Group 4 had higher risk of mortality (aHR: 1.11(95% CI: 1.03–1.19)). Risk of CVD outcomes vary by pattern of remission over time, with lowest risk for those in remission longer. People who achieve remission, even for shorter periods of time, continue to benefit from this lower exposure to hyperglycaemia, which may, in turn, lower the risk of CVD outcomes including mortality.

However, direct data requests can be made to the database Electronic Care and Health Information Analytics (CHIA) governance team, who may be contacted by email: info.chie@nhs.net or phone: +44(0)3001231519."

**Funding:** HDM is an Associate Professor in Primary Care Research and received National Institute for Health Research School of Primary Care Research (NIHR SPCR) funding (SPCR2014-10043) for this project. The views and opinions expressed by authors in this publication are those of the authors and do not necessarily reflect those of the UK National Institute for Health Research (NIHR) or the Department of Health and Social Care. AF is a NIHR Senior Investigator and receives support from NIHR Oxford BioMedical Research Centre. The University of Cambridge has received salary support in respect of SJG from the NHS in the East of England through the Clinical Academic Reserve. The specific role of this author is articulated in the 'author contributions' section. The funders had no role in study design, data collection and analysis, decision to publish, or preparation of the manuscript.

**Competing interests:** The authors have declared that no competing interests exist.

# Introduction

Type 2 diabetes affects 463 million adults globally corresponding to 6.28% of the world's population and total diabetes-related health expenditure is estimated to be over £570 billion [1]. This substantial economic burden is in part related to associated cardiovascular disease (CVD). People with type 2 diabetes compared to those without are more likely to have CVD including peripheral arterial disease, ischaemic stroke, stable angina, heart failure, and non-fatal myocardial infarction. Intensive multifactorial management is effective at reducing these complications, and recent evidence demonstrates that biochemical remission of the disease is achievable through dietary changes, physical activity and subsequent weight loss [2,3]. Remission is defined as a level of glycaemia below the diagnostic threshold ($HbA_{1c} < 6.5\%$ or 48 mmol/mol) in the absence of medication or bariatric surgery. We have previously demonstrated that $\geq 10\%$ weight loss achieved early after diagnosis is strongly associated with remission ((RR 2.43 (95% CI 1.78 to 3.31, p<0.01)) [4]. Remission is often temporary and over the course of type 2 diabetes, individuals will move between states of remission and relapse. To date, these longitudinal patterns of remission have not been described in large population-based cohorts.

It is plausible that since remission is defined by $HbA_{1c}$ level, previous studies on the association between glycaemia and CVD outcomes might be comparable. Observational studies consistently demonstrate a positive association between glycaemia and CVD [5], whereas evaluations of interventions to lower glucose report heterogeneous findings [6–8]. The Look Ahead trial of an intensive behavioural intervention was terminated due to futility in relation to the CVD endpoint [9]. The results of trials of pharmacological interventions to lower glucose have varied according to the drugs (or combination of drugs) undergoing evaluation, the speed and extent of reduction of glucose levels, participants' existing CVD risk and their point in the disease trajectory at baseline [6–8]. Extrapolation of these findings to characterise the impact of remission on the development of CVD outcomes is therefore challenging. To our knowledge, one study has examined the impact of remission on long-term CVD outcomes, with earlier studies focusing on short-term CVD outcomes [4,10–12]. This study reported that remission was associated with lower risk of microvascular complications, macrovascular complications, and CVD events [12]. However, it is unclear whether this risk varies by different patterns of remission over time. This knowledge could inform clinical and policy initiatives which have recently been promoting biochemical remission as a target for management of type 2 diabetes. Accordingly, in this study we describe longitudinal patterns of remission in a large population-based cohort and model these into distinct groups over seven-year follow-up. We then examine risk of CVD outcomes, and all-cause mortality overall and by pattern of remission.

# Materials and methods

## Design

A retrospective cohort study.

## Data source

The Electronic Care and Health Information Analytics (CHIA) database is a pseudo-anonymised live electronic database with routinely collected primary care data for approximately 1.5 million people from 150 primary care practices across Hampshire and the Isle of Wight (Southern England, UK) with linked clinical and biochemistry data from local hospitals.

## Population

We identified a cohort of people with type 2 diabetes using the Quality and Outcomes Framework (QOF) Read code diagnosis. QOF coding is used for NHS administration and financial purposes with high levels of accuracy/completeness [13]. From 120,000 people coded with type 2 diabetes by this criteria on the 1st January 2013, we included 60,287 in our cohort who also had linked and continuous records for seven years until 1st April 2020 (or death) and for whom remission status could be assessed.

## Exposure

Remission was defined as having two $HbA_{1c}$ level < 48 mmol/mol (6.5%) measurements separated over a period of at least six months in the absence of diabetes medications or bariatric surgery [14]. Remission status was assessed for people with $HbA_{1c}$ data for at least two follow-up measurements (i.e., those surviving for at least the first 12 months of follow up).

## Outcomes

i.  Macrovascular complication as a composite of stroke, myocardial infarct (MI) coronary heart disease (CHD), peripheral arterial disease (PAD), or amputation

ii.  Microvascular complications as a composite of peripheral neuropathy, retinopathy, and nephropathy

iii.  CVD events as a composite of MI, amputation, and stroke

iv.  All-cause mortality

We used QoF definitions for peripheral neuropathy, retinopathy, and nephropathy and these were captured using read codes from the primary care record. There was complete data on each outcome measure as a result of the linked data.

## Covariates

**Sociodemographic characteristics.** Baseline data were extracted on age, sex, ethnicity (White, Black, Asian, Mixed and other) and socioeconomic status. This was defined using the 2019 Index of Multiple Deprivation (IMD) quintiles which is a small-area measure of socioeconomic status, ranked nationally and comprises seven domains: income, employment, education/skills/training, health and disability, crime, barriers to housing and services, and living environment) were available. IMD 1 represents the most deprived and IMD 5 represents the least deprived groups.

**Clinical variables.** Baseline comorbidities were defined from diagnostic codes from existing QOF conditions including coronary heart disease, chronic kidney disease, chronic obstructive pulmonary disease (COPD), asthma, cancer, dementia, atrial fibrillation, epilepsy, heart failure, stroke, peripheral vascular disease, hypertension, osteoporosis, osteoarthritis, and depression. Frailty was defined using the electronic frailty Index score. Latest smoking status was extracted at the start of the study (1st January 2013). Weight, body mass index (BMI), systolic and diastolic blood pressure and biochemistry measures (including $HbA_{1c}$ total cholesterol, HDL-cholesterol and eGFR) were taken between January 2013 and April 2020 in six-month intervals, where available. For baseline, we used measures recorded between 1st January 2013-1st April 2013.

**Medication.** Prescribed repeat medication data were extracted from the electronic record at 6-month intervals for the duration of the study period. We used the prescriptions between 1st January 2013-1st April 2013 as the baseline.

## Ethics statement

CHIA is an anonymous National Health Service database and all individuals have consented for collection of their medical records for inclusion in the database (written consent). Ethical and governance approval for this study was obtained from the University of Southampton (ERGO 56127), and Care and Health Information Exchange Information Governance Group (CHIE IGG). All data were fully anonymised prior to the research team gaining access to the data.

## Statistical analysis

We summarised baseline characteristics of the whole cohort. There were missing data on ethnicity (49%) and IMD (0.9%). Ethnicity is frequently missing from routinely collected primary care records and we assigned missing data into the white category in keeping with the local population and previous studies [15]. For weight and $HbA_{1c}$ data that were missing (n = 29678 (49.2%) and n = 30002 (49.8%)), we assumed missing at random and imputed these in a model that included the following non-missing variables; age, sex, diabetes duration, total number of comorbidities at baseline, practice ID, and outcome variables. Data were multiply imputed using Markov Chain Monte Carlo using STATA SE 16.0. We used 10 cycles of imputation. Separate similar imputation models were applied for the remaining biochemistry data. All imputed data after patient death were recoded as missing. With a complete dataset, we summarised participant characteristics stratified by remission status.

We then modelled trajectories of $HbA_{1c}$ level (as a binary measure indicating 48 mmol/mol (6.5%) and above or below 48 mmol/mol) over time and adjusting for remission status at each time point using group-based trajectory modelling in STATA (program developed by Jones and Nagin and based on imputed $HbA_{1c}$ data) [16]. Group-based trajectory models (GBTMs) are mixture models that assume a population is composed of a mixture of distinct subgroups of people who have similar developmental trajectories. A series of unadjusted GBTMs were applied to fit 1 through to 6 group models. The shape of the trajectory was determined by first fitting the trajectory as a cubic function and then reducing the function (to quadratic, slope, or intercept only) if higher polynomials were not statistically significant. The number of trajectories in the model was increased by one and the steps were repeated. Participants are assigned to the group they have highest probability of belonging. We considered participants as belonging to a group if the classification probability was >0.80. The best-fitting model was selected based on 3 criteria: (1) the Bayesian Information Criterion (BIC) (where a lower BIC indicates better fit) (2) the odds of correct classification into each group and (3) the average posterior probabilities of group membership, as a measure of classification quality (>0.80 or greater in all group) [16].The best-fitting model was fitted to each imputed dataset and the classification probabilities from each dataset were saved and averaged to determine group membership. We then used descriptive statistics to compare baseline sociodemographic and clinical characteristics for each remission group. Model F statistics were used to test differences in variables across the groups. Multinomial models (unadjusted and adjusted for age, sex, ethnicity, IMD, baseline weight, diabetes duration, number of comorbidities and clustering within practices) were used to examine the association between weight change categories (no change or weight gain, weight loss ($\leq$ 2.5–5%), ($\leq$5–10%) and ($\geq$10%) from baseline weight) and group membership.

We examine associations between weight change categories and remission groups as previous studies have found an association with overall remission [5,17].

We fitted multivariable-adjusted Cox proportional hazards models to quantify the association between remission at any point during the follow-up for the whole cohort and the incidence of i) macrovascular complications ii) microvascular complications, iii) CVD events, and iv) all-cause mortality). We then constructed the same models with each distinct remission group. People with the event of interest before the start of study were excluded from the respective analysis. Quarter of death rather than exact date of death was available in the database therefore, the mid-point of the quarter of death was used as the date of death in the time to event analyses. For participants with multiple outcome events, we used the time the first event occurred in our time to event analyses. Multivariable models were adjusted based on a priori reasoning for age, sex, ethnicity, IMD, baseline weight, diabetes duration, number of comorbidities and clustering within practices. Finally, we ran a sensitivity analysis to test the robustness of our imputation methods by re-running the cox models and including only those with non-missing (non-imputed) data. A p-value of <0.05 was considered as statistical significance in all analyses.

## Results

### Baseline population characteristics

The cohort included 60,287 people with type 2 diabetes with a mean duration of follow-up of 6.9 years. 7,312 (12.1%) people died during follow up. The mean age of the cohort was 64.6 years, most were male (n = 34,408, 57.1%), white (n = 58,148, 96.5%) and with a mean (SD) duration of diabetes of 8.1 (6.8) years. Baseline characteristics are summarised in Table 1. During the 7-year follow-up period, 11,491 (19.1%) people achieved remission at some point for at least a 6-month period. People who achieved remission compared to those who did not were older (p<0.001), more likely to be female (p<0.001), non-smokers (p<0.001), from a less deprived area (p<0.001) and with a lower baseline weight (p<0.001). Those not included in our study cohort (i.e., those diagnosed with diabetes after 1st January 2013 or with less than seven years continuous data) were younger [mean (SD) 58.1 (14.2)], had shorter diabetes duration [mean (SD) = 5.0 (6.7)] and fewer comorbidities at baseline [mean (SD) 0.9 (1.1)], and slightly higher weight at baseline ([mean (SD) 94.2 (0.2)].

### Characteristics of groups by remission trajectory

The best fitting model identified 4 groups with varying patterns of $HbA_{1c}$ level and remission: Group 1 (8,112 [13.5%]; achieving $HbA_{1c}$ <48 mmol/mol (6.5%) followed by increasing $HbA_{1c}$ levels); Group 2 (6,369 [10.6%]; decreasing $HbA_{1c}$ levels); Group 3 (36,557 [60.6%]; stable high $HbA_{1c}$ levels); Group 4 (9,249 [15.3%]; stable low $HbA_{1c}$ levels (<48mmol/mol or <6.5%)). Fig 1 presents the mean $HbA_{1c}$ levels overtime for each group.

The sociodemographic and clinical characteristics for each remission group are summarised in Table 2 below. There were statistically significant differences in characteristics across the four groups. Individuals in Group 2, who had decreasing $HbA_{1c}$ levels, were mainly older and living in less deprived areas. Those in group 3 who had increasing $HbA_{1c}$ levels were younger, more likely to be male, current smokers, and had a longer duration of diabetes. Those in Group 1 (who achieved $HbA_{1c}$ levels <48mmol/mol (6.5%) but then had increased $HbA_{1c}$ levels) were also younger but had shorter diabetes duration and fewer medications prescribed. Those who had low (<48 mmol/mol (6.5%)) $HbA_{1c}$ levels throughout follow-up (Group 4) were likely to be older, living in less deprived areas, with the shortest duration of diabetes and a lower baseline weight (Table 2). Group 3 was selected as the reference group in regression

**Table 1. Baseline characteristics of the type 2 diabetes cohort within the CHIA database stratified by remission status[¥].**

| | | All (n = 60287) | | Remission[¥] (n = 11335) | | Non-remission[¥] (n = 48,607) |
|---|---|---|---|---|---|---|
| **Sociodemographic** | | | | | | |
| Age, years* | | 64.6 (12.0) | | 66.3 (11.9) | | 64.1 (12.0) |
| Male gender, n (%) | | 34408 (57.1) | | 5992 (52.9) | | 28224 (58.1) |
| Ethnicity, n (%) | | | | | | |
| White | | 58148 (96.5) | | 11047 (97.5) | | 46767 (96.2) |
| Black | | 217 (0.4) | | 35 (0.3) | | 181 (0.4) |
| Asian | | 1514 (2.5) | | 190 (1.7) | | 1317 (2.7) |
| Mixed/Other | | 408 (0.7) | | 63 (0.6) | | 342 (0.7) |
| Socioeconomic Status, n (%) | | | | | | |
| Index of Multiple Deprivation quintile 1 (most deprived) | | 7576 (12.6) | | 1173 (10.3) | | 6363 (13.1) |
| Index of Multiple Deprivation quintile 2 | | 12137 (20.1) | | 2280 (20.1) | | 9788 (20.1) |
| Index of Multiple Deprivation quintile 3 | | 11457 (19.0) | | 2018 (17.8) | | 9375 (19.3) |
| Index of Multiple Deprivation quintile 4 | | 13028 (21.6) | | 2554 (22.5) | | 10398 (21.4) |
| Index of Multiple Deprivation quintile 5 (least deprived) | | 16089 (26.7) | | 3310 (29.2) | | 12683 (26.1) |
| Clinical | | | | | | |
| Diabetes duration, years (n = 60138) | 60138 | 8.1 (6.8) | 11324 | 5.9 (5.3) | 48469 | 8.7 (7.0) |
| Frailty Index (n = 60244) | | 0.2 (0.1) | 11316 | 0.2 (0.1) | 48583 | 0.2 (0.1) |
| Total number baseline comorbidities n (%) | | 1.3 (1.2) | | 1.4 (1.2) | | 1.3 (1.2) |
| Hypertension, n (%) | | 30868 (51.2) | | 5968 (52.7) | | 24721 (50.9) |
| Stroke n (%) | | 2584 (4.3) | | 541 (4.8) | | 2027 (4.2) |
| Myocardial Infarction n (%) | | 4208 (7.0) | | 702 (6.2) | | 3483 (7.2) |
| Amputation n (%) | | 648 (1.1) | | 85 (0.7) | | 560 (1.2) |
| Current smoker, n (%) | | 6559 (10.9) | | 1177 (10.4) | | 5345 (11.0) |
| Weight, kg* | | 90.8 (20.7) | | 88.7 (20.4) | | 91.4 (20.7) |
| BMI, kg/m$^2$* | | 31.5 (6.3) | | 30.9 (6.3) | | 31.7 (6.3) |
| Systolic blood pressure, mmHg* | | 136.1 (15.4) | | 136.2 (15.5) | | 136.1 (15.4) |
| Diastolic blood pressure, mmHg* | | 77.2 (9.4) | | 77.2 (9.4) | | 77.2 (9.4) |
| Total cholesterol, mmol/l* | | 4.6 (1.2) | | 4.7 (1.2) | | 4.5 (1.2) |
| HDL cholesterol, mmol/l* | | 1.2 (0.4) | | 1.3 (0.4) | | 1.2 (0.3) |
| HbA$_{1c}$ level, mmol/mol* | | 60.1 (20.4) | | 59.3 (19.7) | | 60.2 (20.6) |
| eGFR | | 73.1 (17.1) | | 72.6 (17.2) | | 73.2 (17.0) |
| Total number of medications prescribed[#] | | 3.9 (2.4) | | 3.3 (2.4) | | 4.1 (2.4) |
| Anti-hypertensive medication, n (%) | | 32509 (53.9) | | 6071 (53.6) | | 26253 (54.0) |
| Lipid-lowering medication, n (%) | | 40992 (68.0) | | 6903 (60.9) | | 33862 (69.7) |
| Hypoglycaemic medication, n(%) | | 41085 (68.1) | | 4087(36.1) | | 36799 (75.7) |

*Mean (SD). Remission was defined as having two HbA$_{1c}$ < 6.5% (48mmol/mol) readings separated by at least a period of 6 months and no oral hypoglycaemic medication and no history of bariatric surgery

[¥]Estimation sample varies across imputations; minimum number of observations reported. Baseline biochemistry data was defined as the mean of any measurements taken between 1[st] January 2013 and 31[st] March 2013) [#]Medication was defined as being prescribed during the first 6 months of the follow-up year (i.e., Jan-Jul 2013).

models as this was the group with high HbA$_{1c}$ levels throughout the study and therefore a useful comparator to estimate associations with the different remission trajectories. Multinomial regression models indicated that, compared to those who did not change weight or those that gained weight, patients who achieved weight loss of ≥10% at 18 months follow-up were more

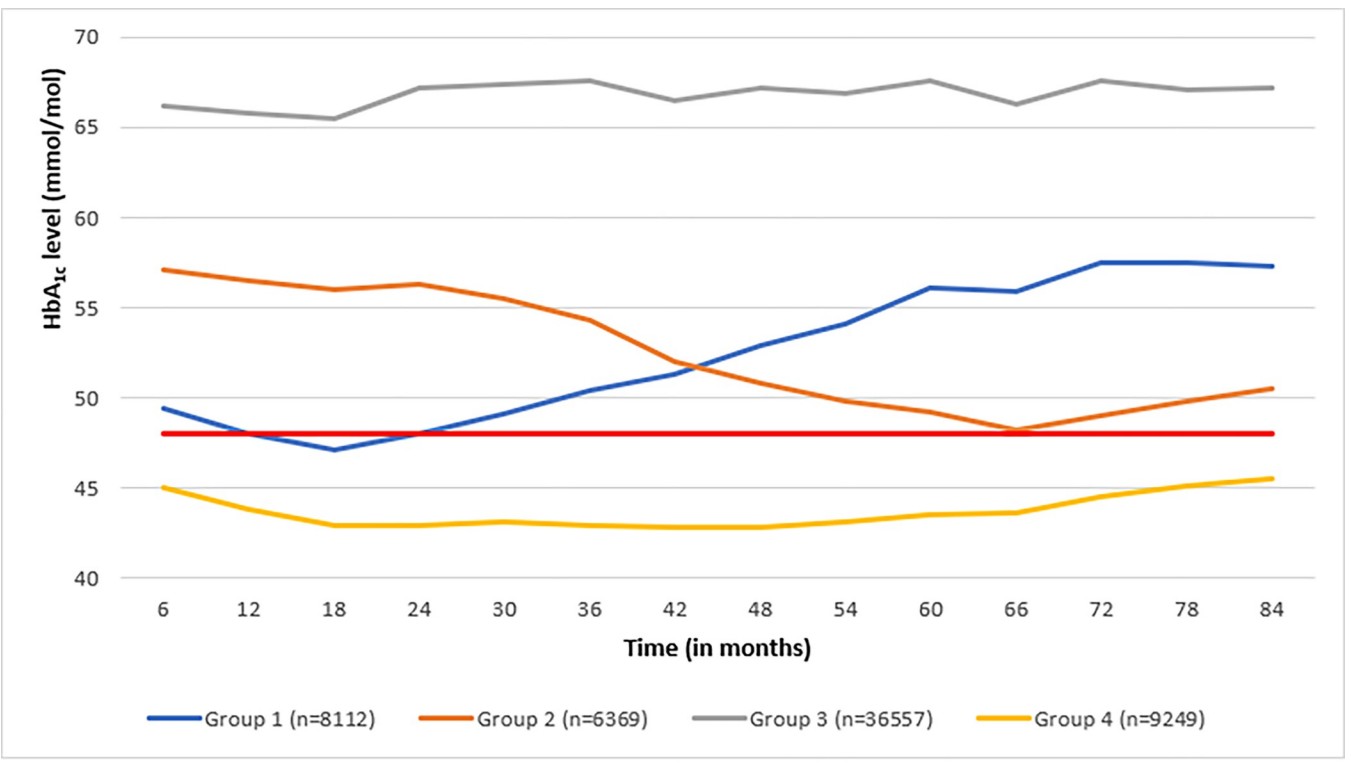

Red line shows the threshold for remission.

**Fig 1. Mean HbA$_{1c}$ level for each remission group over seven-year follow-up within the CHIA type 2 diabetes cohort (n = 60,287).**

likely to be in group 4 compared to group 3 (unadjusted relative risk ratio [RRR] (95% CI): 1.13 (1.06–1.21); adjusted RRR (95% CI): 1.42 (1.31–1.54)) (Table 3). Patients who achieved weight loss of ≥10% at 18 months follow-up were less likely to be in Group 2 compared to Group 3 (unadjusted RRR (95% CI): 0.89 (0.81–0.98); adjusted RRR (95% CI): 0.89 (0.81–0.98). Weight loss of ≥10% at 18 months follow-up was associated with Group 1 (compared to Group 3) in adjusted models but not unadjusted models (Table 3). Weight change at 18 months follow-up was examined as this was the time point at which key differences in HbA$_{1c}$ were noted in Fig 1.

### Remission and CVD outcomes by remission trajectory

In our study cohort, 3,928 (6.5%) had a CVD event, 7,312 (12.1%) died, 4867 (8.1%) people had macrovascular complications, 15,527 (25.8%) had microvascular complications during the study period. In Cox models, people with type 2 diabetes who achieved remission at any point during the seven-year follow-up had a significantly lower risk of CVD events, macrovascular complications and microvascular complications, in both unadjusted and adjusted models. People with type 2 diabetes who achieved remission at any point during the seven-year follow-up had a significantly lower risk of all-cause mortality in adjusted models. These results are shown in S1 Table.

Group 3 (stable high HbA$_{1c}$ levels) was assigned as the reference category for our Cox modelling. Compared to this group, people in all the remaining groups had a lower risk of developing microvascular complication in both unadjusted and adjusted models and those in Groups 1 and 4 also had lower risk of macrovascular complications and CVD events (in

**Table 2. Baseline characteristics by remission group in the CHIA type 2 diabetes cohort (n = 60,287).**

| | Group 1 Achieving HbA$_{1c}$ <48mmol/mol (6.5%) followed by increasing HbA$_{1c}$ levels | Group 2 Decreasing HbA$_{1c}$ levels (>48mmol/mol) | Group 3 Stable high HbA$_{1c}$ levels | Group 4 Stable low HbA$_{1c}$ levels | P-value# |
|---|---|---|---|---|---|
| N (%) | 8112 (13.8) | 6369 (10.7) | 36557 (60.0) | 9249 (15.5) | |
| **Sociodemographic** | | | | | |
| Age, years* | 65.4 (11.6) | 66.8 (11.1) | 63.1 (12.1) | 67.8 (11.3) | 0.419 |
| Male gender, n (%) | 4598 (56.7) | 3609 (56.7) | 21302 (58.3) | 4899 (53.0) | 0.006 |
| Ethnicity, n (%) | | | | | |
| White | 7772 (95.8) | 6132 (96.3) | 35239 (96.4) | 9005 (97.4) | |
| Black | 25 (0.3) | 25 (0.4) | 137 (0.4) | 30 (0.3) | 0.858 |
| Asian | 247 (3.0) | 162 (2.5) | 944 (2.6) | 161 (1.7) | <0.001 |
| Mixed/Other | 68 (0.8) | 50 (0.8) | 237 (0.6) | 53 (0.6) | 0.012 |
| Index of Multiple Deprivation, n (%) | | | | | |
| quintile 1 (most deprived) | 871 (10.8) | 750 (11.8) | 4979 (13.6) | 976 (10.5) | 0.003 |
| quintile 2 | 1575 (19.4) | 1203 (18.9) | 7631 (20.9) | 1727 (18.7) | |
| quintile 3 | 1487 (18.3) | 1151 (18.1) | 7146 (19.5) | 1672 (18.1) | |
| quintile 4 | 1828 (22.5) | 1413 (22.2) | 7702 (21.1) | 2086 (22.5) | |
| quintile 5 (least deprived) | 2351 (29.0) | 1852 (29.1) | 9099 (24.9) | 2788 (30.2) | |
| **Clinical** | | | | | |
| Diabetes duration, years | 6.7 (6.2) | 7.5 (6.1) | 9.1 (7.2) | 6.3 (5.6) | <0.001 |
| Frailty Index | 0.2 (0.1) | 0.2 (0.1) | 0.2 (0.1) | 0.2 (0.1) | 0.043 |
| Total number baseline comorbidities n(%) | 1.3 (1.2) | 1.4 (1.2) | 1.2 (1.2) | 1.4 (1.3) | 0.038 |
| Hypertension, n (%) | 4434 (54.7) | 3573 (56.1) | 177726 (48.6) | 5089 (55.0) | <0.001 |
| Stroke n (%) | 358 (4.4) | 297 (4.7) | 1448 (4.0) | 481 (5.2) | 0.353 |
| Myocardial Infarction n (%) | 518 (6.4) | 457 (7.2) | 2638 (7.2) | 595 (6.4) | 0.537 |
| Amputation n (%) | 67 (0.8) | 79 (1.2) | 428 (1.2) | 74 (0.8) | 0.840 |
| Current smoker, n (%) | 832 (10.3) | 672 (10.6) | 4117 (11.3) | 938 (10.1) | 0.416 |
| Weight, kg* | 89.2 (20.3) | 90.5 (20.7) | 92.1 (20.7) | 87.8 (20.3) | 0.954 |
| BMI, kg/m$^2$* | 31.3 (6.3) | 32.0 (6.4) | 31.7 (6.3) | 30.7 (6.3) | <0.001 |
| Systolic blood pressure, mmHg* | 136.0 (15.2) | 136.9 (15.4) | 136.1 (15.4) | 135.7 (15.5) | 0.128 |
| Diastolic blood pressure, mmHg* | 77.1 (9.4) | 77.0 (9.4) | 77.5 (9.4) | 76.4 (9.3) | 0.044 |
| Total cholesterol, mmol/l* | 4.6 (1.2) | 4.5 (1.2) | 4.6 (1.2) | 4.6 (1.2) | 0.076 |
| HDL cholesterol, mmol/l* | 1.2 (0.4) | 1.2 (0.3) | 1.2 (0.3) | 1.3 (0.4) | <0.001 |
| HbA$_{1c}$ level, mmol/mol* | 59.2 (19.9) | 59.8 (20.1) | 60.5 (20.8) | 59.2 (18.9) | 0.764 |
| eGFR | 72.4 (16.7) | 71.9 (16.9) | 73.8 (17.1) | 71.5 (17.1) | 0.237 |
| Total number of medications prescribed# | 3.8 (2.4) | 4.0 (2.4) | 4.0 (2.4) | 3.8 (2.4) | 0.088 |
| Anti-hypertensive medication, n (%) | 4497 (55.4) | 3717 (58.4) | 18773 (51.4) | 5522 (59.7) | 0.950 |
| Lipid-lowering medication, n (%) | 5512 (67.9) | 4458 (70.0) | 24918 (68.2) | 6104 (66.0) | 0.002 |
| Hypoglycaemic medication, n (%) | 5272 (65.0) | 4337 (68.1) | 26913 (73.6) | 4563 (49.3) | <0.001 |

*Remission group determined for the 60,287 people who were alive at the first follow period (i.e. at 6 months following the start of the study) and therefore had follow-up data. Model F statistics from regression models reported here. Models used on imputed data were linear regression models for continuous variables, logistic regression models for binary variables, and ordered logistic regression models for ordered categorical variables (like IMD).

**Table 3. Multinomial regression showing associations between weight change categories and group membership (compared to group 3).**

| | Unadjusted (n = 59743) | | | | Adjusted[#] (n = 59598) | | | |
|---|---|---|---|---|---|---|---|---|
| | Risk ratio | 95% CI | | p-value | Risk ratio | 95% CI | | p-value |
| **% Weight change category** | | | | | | | | |
| **Group 1** | | | | | | | | |
| No change or weight gain (from baseline) (n = 4360 (53.8%)) | 1.00 | | | | 1.00 | | | |
| Weight loss (≥2.5% to < 5%) (n = 610 (7.5%)) | 1.02 | 0.91 | 1.15 | 0.730 | 1.06 | 0.93 | 1.21 | 0.350 |
| Weight loss (≥5 to <10%) (n = 815 (10.1%)) | 1.04 | 0.92 | 1.17 | 0.531 | 1.11 | 1.00 | 1.24 | 0.047 |
| Weight loss (≥10%) (n = 2322 (28.7%)) | 0.99 | 0.92 | 1.07 | 0.769 | 1.17 | 1.08 | 1.26 | 0.000 |
| **Group 2** | | | | | | | | |
| No change or weight gain (from baseline) (n = 3573 (56.1%)) | 1.00 | | | | 1.00 | | | |
| Weight loss (≥2.5% to < 5%)(n = 470 (7.4%)) | 0.96 | 0.83 | 1.11 | 0.564 | 0.93 | 0.80 | 1.09 | 0.352 |
| Weight loss (≥5 to <10%)(n = 608 (9.6%)) | 0.94 | 0.82 | 1.08 | 0.395 | 0.93 | 0.81 | 1.06 | 0.265 |
| Weight loss (≥10%) (n = 1718 (27.0%)) | 0.89 | 0.81 | 0.98 | 0.017 | 0.89 | 0.81 | 0.98 | 0.017 |
| **Group 4** | | | | | | | | |
| No change or weight gain (from baseline) (n = 4700 (51.2%)) | 1.00 | | | | 1.00 | | | |
| Weight loss (≥2.5% to < 5%)(n = 658 (7.2%)) | 1.02 | 0.90 | 1.16 | 0.733 | 1.08 | 0.94 | 1.25 | 0.254 |
| Weight loss (≥5 to <10%)(n = 954 (10.4%)) | 1.13 | 1.00 | 1.27 | 0.053 | 1.23 | 1.09 | 1.40 | 0.002 |
| Weight loss (≥10%) (n = 2859 (31.2%)) | 1.13 | 1.06 | 1.21 | 0.000 | 1.42 | 1.31 | 1.54 | 0.000 |

[#]Adjusted model includes baseline weight, sociodemographic variables (age, sex, ethnicity and IMD), diabetes duration, number of co-morbidities and clustering within practices.

unadjusted and adjusted models). The risk of these complications was lowest for people in Group 4 who started off and remained at low HbA$_{1c}$ levels (<48mmol/mol or <6.5%) for the entire seven-year follow-up period. Those in Group 1 (who had earlier decrease below HbA$_{1c}$ <48mmol/mol or 6.5%, followed increasing HbA$_{1c}$ levels) also had significantly lower risk of complications compared to Group 3. These results are shown in Table 4.

For risk of all-cause mortality, the rate at which HbA$_{1c}$ levels below 48mmol/mol or 6.5% was achieved was important. Group 2 (decreasing HbA$_{1c}$ levels, though not achieving remission) had similar risk of all-cause mortality to Group 3 (stable high HbA$_{1c}$ levels). Group 1 (achieving HbA$_{1c}$ levels <48 mmol/mol or <6.5% followed by increasing HbA$_{1c}$ levels) had lower risk of all-cause mortality than Group 3. However, Group 4 (i.e., those who started and remained with low HbA$_{1c}$ levels throughout and limited variation) had higher risk of all-cause mortality.

## Discussion

### Main findings

In this population-based cohort of 60,287 people with type 2 diabetes, remission was common with 19% of people achieving remission at some point for at least 6 months. Achieving remission regardless of duration or pattern of HbA$_{1c}$ level and remission status over time was associated with a lower risk of microvascular complications, macrovascular complications, and CVD events. However, the risk of these complications and mortality varied according to remission trajectories over time.

**Table 4. Association between remission group and CVD outcomes in the CHIA type 2 diabetes cohort.**

| | | Unadjusted model | | | | Adjusted model* | | | |
|---|---|---|---|---|---|---|---|---|---|
| | | HR | 95% CI | | p-value | | HR | 95% CI | | p-value |
| **Macrovascular complications** | N = 48,942 | | | | | N = 48,829 | | | | |
| Group 3 (ref) | | 1 | | | | | 1 | | | |
| Group 1 | | 0.85 | 0.78 | 0.93 | <0.001 | | 0.83 | 0.75 | 0.92 | <0.001 |
| Group 2 | | 0.97 | 0.88 | 1.06 | 0.490 | | 0.91 | 0.82 | 1.00 | 0.054 |
| Group 4 | | 0.70 | 0.64 | 0.77 | <0.001 | | 0.66 | 0.61 | 0.71 | <0.001 |
| **Microvascular complications** | N = 41,609 | | | | | N = 41,527 | | | | |
| Group 3 (ref) | | 1 | | | | | 1 | | | |
| Group 1 | | 0.63 | 0.60 | 0.66 | <0.001 | | 0.65 | 0.61 | 0.70 | <0.001 |
| Group 2 | | 0.78 | 0.74 | 0.82 | <0.001 | | 0.80 | 0.76 | 0.85 | <0.001 |
| Group 4 | | 0.56 | 0.53 | 0.59 | <0.001 | | 0.59 | 0.55 | 0.64 | <0.001 |
| **CVD events** | N = 53,218 | | | | | N = 53,097 | | | | |
| Group 3 (ref) | | 1 | | | | | 1 | | | |
| Group 1 | | 0.76 | 0.69 | 0.84 | <0.001 | | 0.74 | 0.67 | 0.83 | <0.001 |
| Group 2 | | 0.94 | 0.85 | 1.04 | 0.254 | | 0.88 | 0.79 | 0.98 | 0.021 |
| Group 4 | | 0.71 | 0.64 | 0.78 | <0.001 | | 0.67 | 0.61 | 0.73 | <0.001 |
| **Death** | N = 60,287 | | | | | N = 60,138 | | | | |
| Group 3 (ref) | | 1 | | | | | 1 | | | |
| Group 1 | | 0.85 | 0.79 | 0.92 | <0.001 | | 0.82 | 0.76 | 0.89 | <0.001 |
| Group 2 | | 1.01 | 0.93 | 1.09 | 0.829 | | 0.92 | 0.85 | 1.01 | 0.086 |
| Group 4 | | 1.29 | 1.21 | 1.37 | <0.001 | | 1.11 | 1.03 | 1.19 | 0.004 |

Group 1 *achieving HbA$_{1c}$ <48 mmol/mol (6.5%) followed by increasing HbA$_{1c}$ levels*); Group 2 *decreasing HbA$_{1c}$ levels*); Group 3 *stable high HbA$_{1c}$ levels*); Group 4 *stable low HbA$_{1c}$ levels (<48mmol/mol or <6.5%)*. The adjusted model shown above included sociodemographic variables (age, sex, ethnicity, IMD), baseline weight, diabetes duration, number of co-morbidities and clustering within practices.

People with event of interest prior to the start of study were excluded from the analysis.

CVD events included a composite of myocardial infarction, amputation and stroke. Microvascular complications included a composite of peripheral neuropathy, retinopathy, and nephropathy. Macrovascular complications include a composite of stroke, MI, coronary heart disease peripheral arterial disease (PAD) and amputation. All-cause mortality was death from any cause.

## Comparison with existing literature

To our knowledge, this is the first study to describe long-term patterns of different type 2 diabetes remission trajectories and their associations with CVD outcomes and all-cause mortality in a population-based cohort. No previous studies have utilised group-based trajectory modelling in this way and instead have examined remission as single whole cohorts [11,12]. Many also include only limited follow-up (<12 months) and therefore have not been able to report on the risk of microvascular complications, macrovascular complication or death [11]. The findings extend our previous findings by highlighting that lower risk of CVD outcomes is achieved regardless of duration of remission, though patients with consistently low HbA$_{1c}$ levels have lowest risk of CVD outcomes. Consistent with the observational studies and in contrast to some of the trials of glucose-lowering drugs [18], we observed consistent trends in unadjusted and adjusted models between remission group and a lower incidence of both macrovascular and microvascular complications. Weight loss of ≥10% was an important predictor of remission trajectory, which is consistent with previous findings on the link between weight loss and remission [4,17].

## Possible explanation for our findings

Although a significant proportion of patients achieved weight loss of $\geq$10% across each group, Group 4 had the highest proportion of patients in this group (31.2%) suggesting that weight loss of $\geq$10% was more likely in this group of people who had lowest levels of $HbA_{1c}$ at baseline. The proportion of patients achieving weight loss of $\geq$10% in Group 2 was lower than the proportion achieving weight loss $\geq$10% in Group 3 (stable high $HbA_{1c}$ levels) which may be unexpected but may be partly due to differences in baseline characteristics such as BMI across the two groups (Table 2) as well as a slightly larger mean decrease in $HbA_{1c}$ level for Group 3 compared to Group 2 (Fig 1); In terms of all-cause mortality, we found that the trajectory of remission was important. Patients in Group 1 (i.e., those achieving remission followed by increasing $HbA_{1c}$ levels) had lower risk of mortality than those in Group 3 (stable high $HbA_{1c}$ levels). This finding suggests that glycaemic control over a period of time may result in improved long-term health outcomes, even following subsequent increases in $HbA_{1c}$. Moreover, Group 1 may have lower risk of mortality due to having consistently lower $HbA_{1c}$ levels compared to Group 3. This reflects increasing risk of mortality at higher levels of $HbA_{1c}$ and is in line with some studies that have reported a linear or J shaped relationship between $HbA_{1c}$ and mortality [19]. Similarly, patients in Group 4 (stable low $HbA_{1c}$ levels with limited variation) had higher risk of mortality. It is possible that people in Group 4 included people who are unwell with a high risk of mortality (such as those with cancer) and therefore could be more likely to lose weight and go into remission. This unintentional weight loss could not be distinguished from intentional weight loss and might be a plausible explanation for some of the observed variations by remission group. Further research is needed to explore which patients achieving remission are at higher risk of mortality. Further work is needed in larger and longer cohorts with more ethnic and socially diverse populations to develop targeted and personalised interventions according to remission group.

## Strengths and limitations

A strength of the study is our large population-based cohort of 60,287 people with type 2 diabetes across a wide geographic region of Southern England including 150 GP practices. The cohort included heterogeneity in age, sex and disease profiles but was limited to mainly people from white ethnicity. This reflects the local area but may not be generalisable to more diverse populations. We included a reasonable follow-up period of seven years with most previous studies examining remission limited to shorter durations [10]. The dataset used is from routinely collected clinical records and is dependent on clinicians accurately recording clinical events and thus is subject to error. However, we used Quality Outcome Framework measures wherever possible which are used for payment and administrative purposes. These measures have previously undergone validity testing and have high levels of completeness and accuracy [13]. We did not have exact dates for deaths in the database with only quarter of death available, so we used the mid-point of the quarter of death in the time to event analyses which may have introduced some error in our analysis. A further limitation was that we were only able to account for prescribed drugs that were captured in the electronic record; we did not have data on whether oral hypoglycaemic drugs were obtained from other sources or the exact date these drugs were prescribed in primary care. Missing data was another limitation of our study which is common with routinely collected data. Although, our sensitivity analysis demonstrated the robustness of our imputation methods as similar associations were observed in the non-imputed analysis. It is possible that some of our findings may be due to chance as we did conduct a number of hypothesis tests. Given that we observed consistent trends across all models and groups, this is less likely. Finally, we cannot rule out reverse causality.

## Conclusions

Remission of type 2 diabetes at any point during the course of diabetes is common in routine clinical care but patterns of remission including maintenance, vary considerably. People who achieve remission, even for shorter periods of time, continue to benefit from this lower exposure to hyperglycaemia, which may, in turn, lower the risk of CVD outcomes including mortality.

## Supporting information

**S1 Table. Association between remission and incidence of CVD outcomes and mortality over seven-year follow in the CHIA type 2 diabetes cohort.**
(DOCX)

## Author Contributions

**Conceptualization:** Beth Stuart, Andrew Farmer, Simon Griffin.

**Data curation:** Hilda O. Hounkpatin.

**Formal analysis:** Hilda O. Hounkpatin.

**Funding acquisition:** Hajira Dambha-Miller, Beth Stuart, Andrew Farmer, Simon Griffin.

**Methodology:** Hilda O. Hounkpatin, Beth Stuart.

**Project administration:** Hajira Dambha-Miller.

**Supervision:** Beth Stuart.

**Writing – original draft:** Hajira Dambha-Miller, Hilda O. Hounkpatin.

**Writing – review & editing:** Hajira Dambha-Miller, Hilda O. Hounkpatin, Beth Stuart, Andrew Farmer, Simon Griffin.

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
