## [Decision Letter · Decision Letter 0]

21 Jun 2023

PONE-D-23-05345Type 2 diabetes remission trajectories and variation in risk of diabetes complications: A population-based cohort studyPLOS ONE

Dear Dr. Hounkpatin,

Thank you for submitting your manuscript to PLOS ONE. After careful consideration, we feel that it has merit but does not fully meet PLOS ONE’s publication criteria as it currently stands. Therefore, we invite you to submit a revised version of the manuscript that addresses the points raised during the review process.

We look forward to receiving your revised manuscript.

Kind regards,

Billy Morara Tsima, MD MSc

Academic Editor

PLOS ONE

Journal Requirements:

2. Please provide additional details regarding participant consent. In the ethics statement in the Methods and online submission information, please ensure that you have specified (1) whether consent was informed and (2) what type you obtained (for instance, written or verbal, and if verbal, how it was documented and witnessed). If your study included minors, state whether you obtained consent from parents or guardians. If the need for consent was waived by the ethics committee, please include this information. If you are reporting a retrospective study of medical records or archived samples, please ensure that you have discussed whether all data were fully anonymized before you accessed them and/or whether the IRB or ethics committee waived the requirement for informed consent. If patients provided informed written consent to have data from their medical records used in research, please include this information.

4. We noted in your submission details that a portion of your manuscript may have been presented or published elsewhere. Please clarify whether this publication was peer-reviewed and formally published. If this work was previously peer-reviewed and published, in the cover letter please provide the reason that this work does not constitute dual publication and should be included in the current manuscript.

Reviewers' comments:

Reviewer's Responses to Questions

**Comments to the Author**

1. Is the manuscript technically sound, and do the data support the conclusions?

Reviewer #1: Yes

Reviewer #2: Partly

2. Has the statistical analysis been performed appropriately and rigorously? 

Reviewer #1: Yes

Reviewer #2: Yes

3. Have the authors made all data underlying the findings in their manuscript fully available?

Reviewer #1: Yes

Reviewer #2: Yes

4. Is the manuscript presented in an intelligible fashion and written in standard English?

Reviewer #1: Yes

Reviewer #2: Yes

5. Review Comments to the Author

Reviewer #1: Abstract

The abstract is well written and easy to read. The fact that the researchers has found that Group 4, which was the group that had a stable low HbA1c levels (<48mmol/mol or <6.5%)), had the highest risk of mortality is a significant finding, which must form part of their conclusion. It does not suffice to say further research is needed to understand which patients achieving remission are at higher risk of mortality. Your conclusion must be guided by your study findings. Common sense may dictate that tight glycemic control is associated with low risk of complications and low risk mortality but the real world maybe different

INTRODUCTION

This not labeled in accordance with accepted journal format. Please label it at the very beginning in bold letters. The introduction is adequate. It has literature review and state the rationale and objectives of the study.

METHODS

The methods and statistical analysis are scientifically sounds. They are satisfactory

RESULTS

The results are well presented in a logical and comprehensive manner. The tables captures important outcomes and they are properly labelled.

DISCUSSIONS

The authors have not really allocated a specific discussion section for this manuscript but rather preferred to include some components of the discussion underneath the CONCLUSIONS. The Researchers has not really allocated much to discuss the findings of their study BUT rather they have chosen to place more emphasis on the strength and limitations of the study.

I will call upon the authors to discuss their findings in details and what they entail. They need to tell us more about the findings of increased mortality in Group 4 and less mortality in Group 1 and come up with possibilities for these findings, even if it maybe hypothetical discussions/reasoning. They may also consider downsizing their Strength/weakness so that they do not exceed the word limit allowed by the journal.

CONCLUSION

The Researchers need to write a brief and concise conclusion that wraps up the FINDINGS of their research.

ACKNOWLEDGEMENTS

I have nothing to add here

REFERENCES

They are adequate and appropriate

Tables and Figures

They EXCELLENT and they improve the readability of the manuscript.

Reviewer #2: In this paper “Type 2 diabetes remission trajectories and variation in risk of diabetes complications: A population-based cohort study”, the authors used Group-Based Trajectory Model to create remission groups and estimate hazard of complications in that groups. To do so, they included a cohort of 60,287 people with type 2 diabetes across a wide geographic region of Southern England. It is valuable work but there are weakness in variables selection, finding presentation, and interpretation.

Abstract:

Please present results without confidence interval in the Abstract, it is recommended give the p-value.

Introduction:

It is not necessary to narrate values of confidence intervals which estimated in other study.

Reference 1, inserted two times in a continues sentence, please correct that.

At first, authors cited reference 3 for saying the effect of interventions and then use that for futility of intervention!

Population, covariates and analysis:

Please report number of missing (and percent) for Weight and HBA1c

Authors must give information about generated imputation.

It is recommended to use SSBIC and CAIC for select better model.

Author explain about uncertainty in assignment to trajectory, how is it considered?

Please determine the significant value in Analysis method.

How authors identify the predictors for trajectory groups?

Please, list the statistical tests used for comparison of variables presented in table 2.

In table 1,

Express all percent for column; it seems that the percent for medications is wrong.

Please check numbers for Comorbidity, Medication prescribed,

Author claim that Remission are older than Non-remission, is there a statistical test for it?

Also, in results, authors say "There were statistically significant differences in characteristics across the four groups". It is recommend that explain about significant variable belong providing p-value.

Weight is a non-significant variable among 4 trajectory groups; please explain about the reason for estimating the risk ratio in different category of Loss weight.

Authors describe about considering loss weight at 18 month of fallow-up, was this only due to change in HBA1c? Authors not provided any information about number of loss weighted in each category by trajectory groups.

Please present number (and percent) for each category of Loss weight variable.

Authors estimate the risk of trajectory group, so it seems that consider group 4 as reference for comprehensibility and adherence.

In ‘Remission and CVD outcomes by remission trajectory’ section;

How set time for subject with multiple complications, please explain that.

Provided descriptions are disagreeable with table, for example there is not significant for G2.

In table 4, number of complications in each trajectory groups is necessary.

In page 10, first paragraph, what is the mean RRR? It seems that author mean was Risk Ratio.

Author must deeply, clinically, and statistically interpretation of all result specifically for table 3 and 4.

In page 17, title for figure repeated two times.

Reference need to be written in Vancouver style.

6. PLOS authors have the option to publish the peer review history of their article (what does this mean?). If published, this will include your full peer review and any attached files.

Reviewer #1: **Yes: **Dr Dipesalema Joel

Reviewer #2: **Yes: **Hssein Ali Adineh

---

## [Author Response · Author response to Decision Letter 0]

24 Jul 2023

**We thank the Editor and Reviewers for their interest in our study and helpful comments

Journal Requirements:

**We have reformatted to the manuscript to meet the journal’s requirements.

2. Please provide additional details regarding participant consent. In the ethics statement in the Methods and online submission information, please ensure that you have specified (1) whether consent was informed and (2) what type you obtained (for instance, written or verbal, and if verbal, how it was documented and witnessed). If your study included minors, state whether you obtained consent from parents or guardians. If the need for consent was waived by the ethics committee, please include this information. If you are reporting a retrospective study of medical records or archived samples, please ensure that you have discussed whether all data were fully anonymized before you accessed them and/or whether the IRB or ethics committee waived the requirement for informed consent. If patients provided informed written consent to have data from their medical records used in research, please include this information.

** Thank you for highlighting this. We have now included an ethics section to the paper (page 7) as below:

“Ethics statement 

CHIA is an anonymous National Health Service database and all individuals have consented for collection of their medical records for inclusion in the database (written consent). Ethical and governance approval for this study was obtained from the University of Southampton (ERGO 56127), and Care and Health Information Exchange Information Governance Group (CHIE IGG). All data were fully anonymised prior to the research team gaining access to the data.”

**Please see response to Q2 above.

4. We noted in your submission details that a portion of your manuscript may have been presented or published elsewhere. Please clarify whether this publication was peer-reviewed and formally published. If this work was previously peer-reviewed and published, in the cover letter please provide the reason that this work does not constitute dual publication and should be included in the current manuscript.

**Thank you for this point. Our earlier publication was peer-reviewed and has been published in Endocrinology, Diabetes & Metabolism . In the cover letter, we have now explained that although both these papers are on a similar subject, they are completely distinct in their aims, methods, analyses, and results although it is the same population being examined. No aspect of the aim, methods or results have been published elsewhere. To clarify:

• The previous publication aimed to examine the relationship between remission and CVD outcomes. The present paper expands on this work to understand the granularity of this association specifically in relation to microvascular and macrovascular complications, as well as CVD events. 

• This current paper also uses different analytical methods including group-based modelling to examine whether achieving remission at different time points over the course of diabetes relates to risk of poor outcomes. Our results find that patients who consistently have low HbA1c levels (<48mmol/mol or 6.5%) have the lowest risk of CVD outcomes, though patients who achieve remission for shorter periods also have a lower risk of CVD outcomes compared to those who do not achieve remission. Additionally, we report baseline characteristics of patients in each group, which clinicians may find helpful to predict trajectories of patients. You will see that the results sections are distinct in each paper. We present and cite some of the results of earlier paper in the current paper for completeness. 

• Below is a table illustrating the differences across the two papers:

 Previous Publication Current paper

Aims To quantify the association between type 2 diabetes remission and 5-year incidence of cardiovascular disease outcomes, overall and in pre-defined subgroups ( age [<45, 45-54,55-64, 75-84, 85+], sex [male, female], diabetes duration [<5, 5-<10, 10-<20, 20+ years), pre-existing CVD (no/yes; defined as a composite of myocardial infarction (MI), amputation, and stroke), baseline BMI [underweight (<18.5kg/m2), normal (18.5-24.9kg/m2), overweight (25-29.9kg/m2), obese (>=30kg/m2)] (13), baseline HbA1c [<6.5%) (48mmol/mol); 6.5-8% (48-63.9 mmol/mol); 8-9% (64-74.9 mmol/mol); >9% (75mmol/mol)] and number of co-morbidities (0, 1-2, 3+) ) To identify distinct diabetes remission trajectories in a large population-based cohort over seven-years follow-up and to examine associations between remission trajectories and diabetes complications

Study sample 60,287 adults (aged 18-85 years) over 7 years who were coded for type 2 diabetes and who had continuously recorded electronic records over seven years from the 1st January 2013 to 1st April 2020 60,287 adults (aged 18-85 years) over 7 years who were coded for type 2 diabetes and who had continuously recorded electronic records over seven years from the 1st January 2013 to 1st April 2020

Study exposure Remission (at any point during the first two years of the study period: 1st January 2013-31st December 2015) Remission trajectory group; Remission at any point during 1st January 2013 to 1st April 2020 (or death)

Study outcome CVD events, microvascular and macrovascular complications All-cause mortality, CVD events, microvascular and macrovascular complications

Study time points Remission during 2013-2015 and CVD outcome between 2016-2020 2013-2020

Methodological Approach Multivariable regression models and Cox regression models Group-based trajectory models, multinomial regression models, Cox regression model

Results Baseline characteristics of overall sample; 

Association between remission and incidence of CVD outcomes over five-year follow-up; 

Size and statistical significance of interaction of remission status (no remission vs remission) with pre-defined subgroups (age, sex, diabetes duration, pre-existing CVD, baseline BMI, baseline HbA1c and number of comorbidities)

Association of above remission-subgroup interactions on CVD events, 

microvascular complications, 

macrovascular complications; 

Association between remission of and incidence of microvascular complications over five-year follow-up by subgroups, Baseline characteristics of overall sample; 

Derivation of remission trajectory groups (based on variation in HbA1c levels over the 7 year follow-up period, additionally adjusting for remission status)

Baseline characteristics by remission trajectory; 

Association between weight change categories and remission trajectory; 

Association between remission-trajectory and CVD outcomes and mortality

Conclusions Achieving remission of type 2 diabetes is associated with a lower risk of microvascular complications, particularly for younger groups and those with fewer comorbidities. Four different trajectories of remission were identified. Risk of CVD outcomes vary by pattern of remission over time, with lowest risk for those in remission longer. People who achieve remission, even for shorter periods of time, continue to benefit from this lower exposure to hyperglycaemia, which may, in turn, lower the risk of CVD outcomes including mortality

** Thank you. We do not have governance permissions to share individual-level data on which these analyses were conducted since they derive from clinical record data. However, direct data requests can be made to the database Electronic Care and Health Information Analytics (CHIA) governance team.

**Thank you.

** We have now added this to page 7 of the manuscript.

** We have now added this to page 24 of the manuscript.

**We have checked and updated the reference list to meet the referencing style in the guidelines. References 1 and 8 have been changed as previous references (as shown in tracked changes) could no longer be accessed. 

Reviewers' comments:

Reviewer #1: Abstract

The abstract is well written and easy to read. The fact that the researchers has found that Group 4, which was the group that had a stable low HbA1c levels (<48mmol/mol or <6.5%)), had the highest risk of mortality is a significant finding, which must form part of their conclusion. It does not suffice to say further research is needed to understand which patients achieving remission are at higher risk of mortality. Your conclusion must be guided by your study findings. Common sense may dictate that tight glycemic control is associated with low risk of complications and low risk mortality but the real world maybe different

**Thank you for this important point. We agree and have now concluded that “risk of CVD outcomes vary by pattern of remission over time, with lowest risk for those in remission longer. People who achieve remission, even for shorter periods of time, continue to benefit from this lower exposure to hyperglycaemia, which may, in turn, lower the risk of CVD outcomes including mortality.”

INTRODUCTION

This not labeled in accordance with accepted journal format. Please label it at the very beginning in bold letters. The introduction is adequate. It has literature review and state the rationale and objectives of the study.

**This has now been corrected and appropriate labels are now used throughout the manuscript. 

METHODS

The methods and statistical analysis are scientifically sounds. They are satisfactory

**Thank you.

RESULTS

The results are well presented in a logical and comprehensive manner. The tables captures important outcomes and they are properly labelled.

**Thank you.

DISCUSSIONS

The authors have not really allocated a specific discussion section for this manuscript but rather preferred to include some components of the discussion underneath the CONCLUSIONS. The Researchers has not really allocated much to discuss the findings of their study BUT rather they have chosen to place more emphasis on the strength and limitations of the study.

I will call upon the authors to discuss their findings in details and what they entail. They need to tell us more about the findings of increased mortality in Group 4 and less mortality in Group 1 and come up with possibilities for these findings, even if it maybe hypothetical discussions/reasoning. They may also consider downsizing their Strength/weakness so that they do not exceed the word limit allowed by the journal.

**Thank you for this very helpful suggestion. We have now added a detailed discussion section to the paper which more fully discusses the findings of our study, particularly Table 3 and 4. Please see the discussion section on pages 19-22, which includes a section on ‘possible explanations for our findings’.

CONCLUSION

The Researchers need to write a brief and concise conclusion that wraps up the FINDINGS of their research.

**We have revised the conclusion so it is now succinct and focused on our study findings. Please see page 22.

ACKNOWLEDGEMENTS

I have nothing to add here

REFERENCES

They are adequate and appropriate

Tables and Figures

They EXCELLENT and they improve the readability of the manuscript.

**We thank the Reviewer for their helpful and encouraging comments and suggestions.

Reviewer #2: In this paper “Type 2 diabetes remission trajectories and variation in risk of diabetes complications: A population-based cohort study”, the authors used Group-Based Trajectory Model to create remission groups and estimate hazard of complications in that groups. To do so, they included a cohort of 60,287 people with type 2 diabetes across a wide geographic region of Southern England. It is valuable work but there are weakness in variables selection, finding presentation, and interpretation.

Abstract:

Please present results without confidence interval in the Abstract, it is recommended give the p-value.

**Thank you for this helpful comment. We feel it is helpful to have the confidence intervals and some readers may be interested in this. However, as the Reviewer suggests, we have also included the p-values in the abstract. 

Introduction:

It is not necessary to narrate values of confidence intervals which estimated in other study.

**We agree and have now removed these from the introduction.

Reference 1, inserted two times in a continues sentence, please correct that.

**We have now corrected this. 

At first, authors cited reference 3 for saying the effect of interventions and then use that for futility of intervention!

**Thank you for pointing this out. This has now been corrected.

Population, covariates and analysis:

Please report number of missing (and percent) for Weight and HBA1c

**We have now stated this on page 8 line 5. 

Authors must give information about generated imputation.

**All data presented in the tables and text here is based on the generated imputation. We describe how data was imputed in the methods section (1st paragraph on page 8).

It is recommended to use SSBIC and CAIC for select better model.

**Thank you for this helpful suggestion. We have used the sample-size BIC (SSBIC) here as well as other criterion as these are the criteria that have been suggested to use for group-based trajectory modelling methodology as cited in the reference 16 and this is the methodology we have followed. The SSBIC considers the likelihood of the model as well as the number of parameters in the model. However, it is very often the case that both the SSBIC and cAIC select the same model.

Author explain about uncertainty in assignment to trajectory, how is it considered?

**Thank you for highlighting this. Uncertainty is always a possibility but reduced by the use of multiple selection criteria (Mesidor et al, 2022). Participants are assigned to the group they have highest probability of belonging. We considered participants as belonging to a group if the classification probability was >0.80. 

Mésidor M, Rousseau MC, O'Loughlin J, Sylvestre MP. Does group-based trajectory modeling estimate spurious trajectories? BMC Med Res Methodol. 2022;22(1):194. doi: 10.1186/s12874-022-01622-9.

Please determine the significant value in Analysis method.

**We have included, on page 9, that a p-value of <0.05 was considered as statistical significance in all analyses.

How authors identify the predictors for trajectory groups?

**Predictors of trajectory group membership were assessed with the multivariable regression models. Potential predictors were selected (a priori) and included in these models based on the existing literature and discussion with the research team which included clinical academics.

Please, list the statistical tests used for comparison of variables presented in table 2.

**We report Model F statistics for all test in Table 2. This is because we use multiple imputed data which are better suited to statistical models rather than statistical tests. Regression models were used for continuous variables, logistic regression models for binary variables, and ordered logistic regression models for ordered categorical variables (like IMD). We report this on the footnote of the Table 2 on page 15.

In table 1,

Express all percent for column; it seems that the percent for medications is wrong. 

**Thank you for this point and apologies for the confusion. As this is a large table, for comorbidities, smoking status, and prescribed medication we have decided to present the number and percentage for those that have, for example, the comorbidity. We would like to keep the table as it is (to avoid having a very large table), unless the journal requires us to change this.

Please check numbers for Comorbidity, Medication prescribed,

**We have checked the numbers and it is correct. 

Author claim that Remission are older than Non-remission, is there a statistical test for it?

**Thanks for highlighting this. We have now fitted appropriate regression models (as described above) and these differences are statistically significant. We report this on lines 7-8 on page 10.

Also, in results, authors say "There were statistically significant differences in characteristics across the four groups". It is recommend that explain about significant variable belong providing p-value.

**Thank you for this point. In the text, we refer the reader to Table 2 where we present the p-values for each variable. Table 2 shows gender, IMD, ethnicity, diabetes duration, frailty, and baseline comorbidities amongst others vary across the groups. 

Weight is a non-significant variable among 4 trajectory groups; please explain about the reason for estimating the risk ratio in different category of Loss weight.

**This is an important point. The Reviewer is right that there was no significant association between trajectory and weight change (as a continuous variable). However, we know from the literature as well as our previous work that weight change is important for remission and significant associations with remission have been shown using weight change categories (reference 6 and 17 in the paper) We have justified this on lines 11-12 on page 9.

Authors describe about considering loss weight at 18 month of fallow-up, was this only due to change in HBA1c? Authors not provided any information about number of loss weighted in each category by trajectory groups.Please present number (and percent) for each category of Loss weight variable.

**18 month follow up point was selected based on Figure 1 (graph showing mean HbA1c level over time for each remission group). The graph shows a decrease in HbA1c level for all groups and the largest differences in mean HbA1c level across groups at 18 months.

**We have now presented number and percentages to Table 3.

Authors estimate the risk of trajectory group, so it seems that consider group 4 as reference for comprehensibility and adherence.

**Thank you. Using Group 4 (those that remain in remission throughout study) would not allow us to estimate associations with stable remission. Therefore, we have decided to use Group 3 (the group that have high HbA1c throughout the study period) as the reference so that associations with different degrees of remission can be estimated compared to this group (never achieving remission). This is also consistent with the analyses assessing associations with remission at any time point, where ‘no remission’ is the reference (S1 Table). We have added this to lines 7-9 of page 13.

In ‘Remission and CVD outcomes by remission trajectory’ section;

How set time for subject with multiple complications, please explain that.

**CVD event, microvascular complication and macrovascular complications were each composite measures. For subjects with multiple events, we defined study period as the start of the study until the time the first event occurred. We have added this to the ‘Statistical analysis’ section on page 9. We have used a maximum follow-up of 7 years for all study participants.

Provided descriptions are disagreeable with table, for example there is not significant for G2.

**We have checked Table 3 and Table 3 shows significant association of weight loss of ≥10% with Group 2 membership RRR: 0.89(0.81-0.98), p-value=0.017, both in unadjusted and adjusted models. We have further discussed this finding in the discussion section on page 20.

In table 4, number of complications in each trajectory groups is necessary.

**Thank you for this point. We did not look at the association between number of complications and trajectories as this was not the aim of the study and may also require longer follow-up period than was available to us. Further, Table 4 reports on associations using Cox models and results would not be directly comparable with a regression model more suited to a continuous outcome (number of complications). 

In page 10, first paragraph, what is the mean RRR? It seems that author mean was Risk Ratio.

**This means Relative Risk Ratio as it is a multinomial model. We have now defined on first use.

Author must deeply, clinically, and statistically interpretation of all result specifically for table 3 and 4.

** Thank you for this very helpful suggestion. We have now added a detailed discussion section to the paper which more fully discusses the findings of our study, particularly Table 3 and 4. Please see the discussion section on pages 19-22, which includes a section on ‘possible explanations for our findings’.

In page 17, title for figure repeated two times.

**This has now been corrected.

Reference need to be written in Vancouver style.

**Thank you. We have updated to the reference list to Vancouver style.

---

## [Editor Report · Decision Letter 1]

16 Aug 2023

Type 2 diabetes remission trajectories and variation in risk of diabetes complications: A population-based cohort study

PONE-D-23-05345R1

Dear Dr. Hounkpatin,

We’re pleased to inform you that your manuscript has been judged scientifically suitable for publication and will be formally accepted for publication once it meets all outstanding technical requirements.

Kind regards,

Billy Morara Tsima, MD MSc

Academic Editor

PLOS ONE
---

## [Editor Report · Acceptance letter]

21 Aug 2023

PONE-D-23-05345R1 

Type 2 diabetes remission trajectories and variation in risk of diabetes complications: A population-based cohort study 

Dear Dr. Hounkpatin:

I'm pleased to inform you that your manuscript has been deemed suitable for publication in PLOS ONE. Congratulations! Your manuscript is now with our production department. 

Kind regards, 

on behalf of

Dr. Billy Morara Tsima 

Academic Editor

PLOS ONE